# Demographic-environmental effect on dengue outbreaks in 11 countries

**Anamul Haque Sajib**[1], **Sabina Akter**[1], **Goutam Saha**[2,3]*, **Zakir Hossain**[1]

**1** Department of Statistics, University of Dhaka, Dhaka, Bangladesh, **2** Department of Mathematics, University of Dhaka, Dhaka, Bangladesh, **3** Miyan Research Institute, International University of Business Agriculture and Technology, Uttara, Dhaka, Bangladesh

\* gsahamath@du.ac.bd

**Data Availability Statement:** Data availability statement: All the data is deposited in our repository and freely available at https://doi.org/10.6084/m9.figshare.25745295.v1.

## Abstract

### Background

Dengue outbreaks are common in tropical or temperate countries, and climate change can exacerbate the problem by creating conditions conducive to the spread of mosquitoes and prolonging the transmission season. Warmer temperatures can allow mosquitoes to mature faster and increase their ability to spread disease. Additionally, changes in rainfall patterns can create more standing water, providing a breeding ground for mosquitoes.

### Objective

The objective of this study is to investigate the correlation between environmental and demographic factors and the dissemination of dengue fever. The study will use yearly data from 2000 to 2021 from 11 countries highly affected by dengue, considering multiple factors such as dengue cases, temperatures, precipitation, and population to better understand the impact of these variables on dengue transmission.

### Methods

In this research, Poisson regression (PR) and negative binomial regression (NBR) models are used to model count data and estimate the effect of different predictor variables on the outcome. Also, histogram plots and pairwise correlation plots are used to provide an initial overview of the distribution and relationship between the variables. Moreover, Goodness-of-fit tests, t-test analysis, diagnostic plots, influence plots, and residual vs. leverage plots are used to check the assumptions and validity of the models and identify any outliers or influential observations that may be affecting the results.

### Results

The findings indicate that mean temperature and log(Urban) had a positive impact on dengue infection rates, while maximum temperature, log(Precipitation), and population density had a negative impact. However, minimum temperature, log(Rural), and log(Total population) did not demonstrate any significant effects on the incidence of dengue.

**Funding:** The author(s) received no specific funding for this work.

**Competing interests:** The authors have declared that no competing interests exist.

## Conclusion

The impact of demographic-environmental factors on dengue outbreaks in 11 Asian countries is illuminated by this study. The results highlight the significance of mean temperature (Tmean), maximum temperature (Tmax), log(Urban), log(Precipitation), and population density in influencing dengue incidence rates. However, further research is needed to gain a better understanding of the role of additional variables, such as immunity levels, awareness, and vector control measures, in the spread of dengue.

## Introduction

Dengue virus is responsible for causing dengue fever, a viral illness transmitted to humans when an Aedes mosquito carrying the virus bites them. Symptoms of dengue fever usually include fever, headache, muscle pain, joint pain, and a distinct skin rash. In severe cases, dengue fever can progress to a life-threatening form called dengue hemorrhagic fever. With an estimated 390 million cases occurring annually, dengue fever is a significant global health concern that predominantly affects tropical and subtropical regions worldwide, with urban and semi-urban areas being at higher risk. The World Health Organization (WHO) has identified dengue fever as a primary cause of illness and mortality in some Asian and Latin American countries. Climate change, population growth, and increased travel and trade have all contributed to the spread of the disease in recent years. Moreover, Dengue and climate change are closely linked, as the mosquitoes that transmit the dengue virus thrive in warm and humid environments. As the global temperature increases, the range of these mosquitoes is expected to expand, leading to a higher risk of dengue transmission in new areas. Climate change also affects precipitation patterns and can lead to increased flooding and standing water, which provide ideal breeding grounds for mosquitoes. In addition, extreme weather events such as cyclones and hurricanes can disrupt public health infrastructure and vector control measures, making it harder to prevent and control outbreaks of dengue. Furthermore, Climate change can also impact human populations in ways that increase their risk of infection, such as population displacement, economic stress, and reduced access to health care. WHO has acknowledged the potential influence of climate change on the spread of vector-borne illnesses, including dengue, and has urged heightened endeavors to adjust to and alleviate these consequences.

Banu et al. [1] used data from the years 2000–2010 to project future dengue outbreaks in Dhaka, Bangladesh. They found a strong association between temperature and humidity with dengue outbreaks. They also projected that an increase in temperature of 3.3 ˚C by 2100 would result in a significant increase in dengue cases. Dhewantara et al. [2] used data from the Ministry of Health, Indonesia from 2012–2017 to study the ecological factors affecting dengue incidence in Bali. They found that rainfall, elevation, and population density were associated with dengue transmission in Bali. Jácome et al. [3] investigated the association of meteorological variables with dengue fever in Guayaquil, Ecuador in 2012. They found that precipitation, maximum and minimum temperatures, solar radiation, wind velocity, and wind vapor pressure were all associated with dengue outbreaks. Jayaraj et al. [4] developed a model using weather and dengue incidence data to predict future dengue incidence in Tawau district, Malaysia, using data collected from 2006 to 2016. They observed that an outbreak of dengue transmission is associated with mean temperature, minimum temperature, and humidity.

They also found that rainfall is not a significant factor for dengue outbreaks which is unusual as rainfall is known to be helpful for mosquito breeding sites. Xu et al. [5] examined the imported dengue cases from two Indonesian cities, Bali and Jakarta, to Australia. They used data collected from Bali (2007–2013) and Jakarta (2001–2017) and found that an increase in dengue cases is associated with an increase in mean temperature and rainfall.

Liu et al. [6] investigated the relationship between climate factors and dengue outbreaks in China using data from 1980–2016. They found that population density, mean temperature, precipitation, and humidity had a positive association with dengue outbreaks. Prabodanie et al. [7] examined the relationship between climate factors and dengue incidence dynamics in the dry and wet zones of Colombo & Batticaloa, Sri Lanka using data collected from 2009–2017. They found that temperature had no relation with dengue incidence in both the dry and wet zones. However, population density, rainfall, and humidity showed an association with dengue spread. Akter et al. [8] employed Bayesian spatial analysis to investigate the correlation between weather, socio-ecological factors, and dengue occurrence in Queensland, Australia. The data was gathered from 2010–2015. They found that average rainfall and an increase in terrace houses were significantly associated with an increase in dengue incidence. Cheng et al. [9] used a Bayesian spatial model to investigate the association between climate variables and dengue incidence in Guangdong, China. They found that extremely high temperatures and rainfall have a strong positive association with dengue incidence. However, they also found that the spatial variability of extremely high temperatures did not show any association with dengue outbreaks, while the opposite behavior was observed for extremely high rainfall. Davis et al. [10] proposed a Suitable Conditions Index (SCI) to predict future dengue forecasts in six Asia-Pacific countries. They used data collected from 2005 to 2018. They presented projected scenarios for 2030, 2050, and 2070 and suggested that the South East Asia (SEA) region will be most suitable for future dengue outbreaks, while Australia and China will be less suitable for mosquitoes breeding due to the less favorable conditions. This is an alarming finding for the SEA region as the socioeconomic and environmental factors and global warming effects are likely to have a severe impact on dengue outbreaks.

Haider et al. [11] discussed the importance of mosquito control in reducing dengue outbreaks in Bangladesh. They used data collected from 2008 to 2019 and found that temperature and rainfall are strongly associated with the growth rate of dengue cases. The authors suggest that controlling mosquitoes, which are vectors of dengue, can be an effective way to reduce the incidence of dengue outbreaks. Muñoz et al. [12] investigated the relationship between local climates, ENSO (El Niño Southern Oscillation), and dengue outbreaks in Columbia. The data collection period was from 2007 to 2017. They found that there is a strong correlation between dengue transmission, maximum temperature, and wind velocity. They also found a weak correlation between dengue outbreaks, humidity, and precipitation. This suggests that while temperature and wind velocity are key factors affecting dengue transmission, humidity and precipitation may have less of an impact. Seah et al. [13] investigated the relationship between heat waves and maximum temperature on dengue incidence in Singapore. The data collection period was from 2009 to 2018. They found that dengue infection reduced as heat and temperature increased, and temperature equal to 31 ˚C or higher made a significant impact on reducing dengue spread. This suggests that high temperatures may be an effective way to reduce dengue transmission in Singapore.

Wang et al. [14] examined the effect of extreme temperature and rainfall on dengue infection in four Asian countries. They used data collected from 2012 to 2020. They found that extreme rainfall helps to reduce the spread of dengue whereas extreme temperature is associated with an increase in dengue infection. This suggests that while extreme rainfall may be beneficial in reducing dengue transmission, extreme temperatures may exacerbate dengue

infection in these countries. Susilawaty et al. [15] investigated the relationship between weather variables and dengue incidence in Makassar, Indonesia. They used data collected from 2011 to 2017. They found that humidity had a strong relationship with dengue incidence, while temperature, wind velocity, and rainfall had a negative correlation with dengue incidence. This suggests that while humidity may be a key factor in dengue transmission in Makassar, temperature, wind velocity, and rainfall may be less important. Costa et al. [16] investigated the relationship between hydroclimatic variables and dengue outbreaks in dryland regions of Ceara, Brazil. The data collection period was from 2008 to 2018. They found that precipitation was not associated with dengue incidence, however, population density and extreme drought were strongly associated with dengue incidence. This suggests that while precipitation may not be a major factor in dengue transmission in dryland regions of Ceara, population density and extreme drought may play a significant role in dengue outbreaks.

While numerous studies have investigated the association between climate variables and dengue transmission, there is a noticeable gap in research that integrates demographic and environmental factors into the analysis, particularly in highly affected countries. Despite many literature on climate-dengue relationships, there remains a need for comprehensive studies that consider the relation between socioeconomic variables, environmental conditions, and dengue spread. Considering the research gap, this research aims to understand the relationship between demographic-environmental factors and the spread of dengue transmission, with a focus on countries that are highly affected by dengue. The use of statistical models and graphical techniques allows for a comprehensive analysis of the data to identify the most influential factors and determine the strength of their relationship with dengue transmission. This study holds significant importance, firstly, by examining the relationship between demographic-environmental factors and dengue transmission in highly affected countries, it addresses a critical gap in the existing literature, providing insights of dengue outbreaks. Secondly, the utilization of statistical models and graphical techniques enables a comprehensive analysis of the data, facilitating the identification of key factors and their importance in driving dengue transmission.

## Data descriptions

This study is collecting a variety of data from 11 different countries that are considered to have high rates of dengue transmission. The data includes yearly dengue incidence, temperature (mean, minimum, and maximum), precipitation, total population, urban and rural population, and population density. This study is collecting data from Bangladesh, Nepal, Sri Lanka, India, Brazil, Indonesia, Malaysia, Philippines, Thailand, Myanmar, and Vietnam. The selection of these countries for data collection in this study is justified by their significant burden of dengue fever, diverse geographical landscapes, and varying socioeconomic conditions. Moreover, the inclusion of these countries allows for the exploration of diverse climatic zones and population densities, thereby enabling a clear understanding of the contextual factors influencing dengue spread. The data collection period is from 2000 to 2021, which spans over two decades. The study is using two sources for data collection, Climate Change Knowledge Portal (https://climateknowledgeportal.worldbank.org/) for climate-related data, and the World Bank website (https://www.worldbank.org/en/home) for population-related data. In addition to collecting data from the two sources previously mentioned, the study is also collecting the total number of yearly dengue cases for each country from the Ministry of Health of the corresponding countries, as well as from an article by Gan et al. [17]. Statistical analysis is being performed using R on the data from all 11 countries. The data is openly available in the study's repository for anyone to access. Link: https://doi.org/10.6084/m9.figshare.25745295.v1

## Methodology

The response variable considered in our study is the number of dengue cases which is a count (discrete) variable in nature. We aim to explore how this response variable is affected by demographic and environmental factors, and this type of relationship can be statistically examined through several count regression models, most importantly Poisson, Negative binomial, zero inflated Poisson and zero inflated Negative binomial models under generalized linear model framework [18]. For a particular data set, suitable count regression models are selected based on the relationship of response variable mean and variance. For example, (i) Poisson regression can be used when mean and variance of the count response are equal, (ii) Negative binomial regression can be used in over dispersed count data (variance of the count response is larger than its mean), (iii) Zero inflated Poisson can be used when response consists of excessive zero count compare to the other counts but its mean and variance are the same and (iv) Zero inflated negative binomial can be used in case of excessive zero counts with over-dispersion. The response variable in our data set is over dispersed as its variance is 485002 times compared to its mean and no excessive zero (even other numbers) count is observed in the data. Therefore, Negative binomial regression model is chosen as a suitable count regression model to analyze the data considered in this paper. We also consider here Poisson regression model as a basic count regression model. PR model presumes that the response variable conforms to a Poisson distribution and establishes the average of the response variable as a function of explanatory variables. NBR model on the other hand, is used when the response variable is over-dispersed relative to a Poisson distribution and models the mean of the response variable as a function of predictor variables. Both models can be useful in understanding the relationship between predictor variables and the count of occurrences of an event.

Assumptions made in the PR model include that the responses $Y_i$, $i = 1, 2, 3, \cdots, n$, are generated from the Poisson distribution with mean $\mu_i$ that is

$$Y_i | \mu_i \sim Poisson(\mu_i) = \frac{e^{-\mu_i} \mu_i^{y_i}}{y_i!} \tag{1}$$

and the $\mu_i$ are the functions of covariates. For the Poisson regression, the bridge between $\mu_i$ and the set of covariates can be defined as

$$\log(\mu_i) = g(\mu_i) = \eta_i = \beta_0 + \beta_{1i} x_{1i} + \cdots\cdots + \beta_{pi} x_{pi} = \beta_0 + \sum_{k=1}^{p} \beta_{ki} x_{ki} \tag{2}$$

where $g$ and $\eta_i$ are the log-link functions and $i^{th}$ linear predictor, respectively while $\beta_0, \beta_1, \cdots, \beta_p$ are the regression coefficients.

One of the common phenomena for modeling count response data is overdispersion which arises due to the excess variation between response counts. The response variable, dengue-infected cases, considered in this study have excess variation. It also arises when the independence assumption of observations is violated and this can happen in clustered data. Standard errors of the estimates are underestimated due to the consequence of overdispersion which makes a variable to be a significant predictor when it is not actually significant. Therefore, overdispersion needs to be taken care of properly in analyzing count response.

Hilbe [19] suggested Pearson $\chi^2$ statistic to detect overdispersion in a real data set which is defined as

$$\chi^2 = \sum_{i=1}^{n} \frac{(y_i - \hat{\mu}_i)^2}{Var(\hat{\mu}_i)} \tag{3}$$

where $Var(\hat{\mu}_i)$ is the variance of expected count $\hat{\mu}_i$. A model is considered to be overdispersed if the dispersion value, $\chi^2$ statistic divided by the corresponding degrees of freedom, is greater than 1 while equidispersed and under-dispersed are considered for the dispersion value 1 and less than 1, respectively. A model that can be used to analyze overdispersed count responses is NBR model. Assumptions made in NBR model include that the responses $Y_i$ are generated from the negative-binomial distribution with the following form

$$Y_i|\mu_i \sim NB(\mu_i, s) = \binom{s + y_i - 1}{y_i} \left(\frac{s}{s + \mu_i}\right)^s \left(\frac{\mu_i}{s + \mu_i}\right)^{y_i} \tag{4}$$

where $\mu_i$ and $s$ are known as the mean and dispersion parameters respectively. The variance of NBR response variable is a quadratic function of its mean which is

$$\mu_i + \frac{\mu_i^2}{s}. \tag{5}$$

NBR model is utilized to model count data that exhibit overdispersion, owing to this attribute. The negative binomial distribution approximates the Poisson distribution as a limiting case when $s$ is large enough. Under NBR framework, the mean $\mu_i$ is also connected with covariates via a log-link function like PR i.e.

$$E(Y_i|x_i) = \mu_i = \exp\left(\beta_0 + \sum_{k=1}^{p} \beta_{ki} x_{ki}\right) \tag{6}$$

In the context of our research, the mean number of dengue cases $\mu_i$ are connected with covariates under both the PR and NBR frameworks as

$$\log(\mu_i) = \eta_i = \beta_0 + \beta_{1i}(Mean\_Tem)_i + \beta_{2i}(Min\_Tem)_i + \beta_{3i}(Max\_Tem)_i$$

$$+ \beta_{4i}(\log(Precipitation))_i + \beta_{5i}(\log(Total\ population))_i$$

$$\tag{7}$$

$$+ \beta_{6i}(\log(Urban))_i + \beta_{7i}(\log(Rural))_i + \beta_{8i}(Population\ density)_i$$

Some of the covariates considered in our study like precipitation, total population, unban population and rural population have very higher values compared to the other covariates values such as mean temperature, min temperature, max temperature and population density. More specifically, the numerical values of mean temperature, min temperature and max temperature are limited to two digits before decimal while population density has three digits value. On the other hand, total population, urban population and rural population have values in 9 digits form while precipitation has 4 digits value. Therefore, scaling of the independent variable values differs drastically which causes numerical instability in the estimation process of the count regression model. We considered log transformation of precipitation, total population, unban population and rural population to make their values similar to other independent variables values which provides numerical stability in the estimation process.

The functions `glm` and `glm.nb` in R under `MASS` package are used to fit Poisson and negative binomial regression models, respectively. We use AIC, dispersion, deviance and Pseudo-$R^2$ values to assess the performance of the fitted model. Akaike [20] developed a criteria, known as Akaike information criterion (AIC), to assess the quality of fit of a model which can be defined as

$$AIC = -2l(\hat{\theta}, y) + 2p \tag{8}$$

where $l\left(\hat{\theta}, y\right)$ is the log-likelihood function at $\theta = \hat{\theta}$, $y$ is the data vector and $p$ is the number of regression coefficients. A model is considered to be the best model provided that its AIC value is the smallest. Models from different GLM families can also be compared using deviance test, and it can be defined as

$$D = 2 \sum_{i=1}^{n} \left\{ l(y, y) - l\left(\hat{\theta}, y\right) \right\} \sim \chi^2_{(n-p)} \qquad (9)$$

where $l(y, y)$ and $l\left(\hat{\theta}, y\right)$ are the log-likelihood functions with $\theta$ is replaced by $y$ and $\hat{\theta}$, respectively. Like the AIC criteria, a model with smaller deviance is considered to be a better model over a model with greater deviance. The comparative deviance statistic D follows a $\chi^2$ distribution with $(n - p)$ degrees of freedom. Finally, the coefficient of determination, $R^2$ is a well-known model evaluating criteria used in linear regression model. However, for a non-linear model like PR, NBR, and logistic regression models $R^2$ is not appropriate, and the counterpart of $R^2$ in such case is pseudo-$R^2$ which is defined as

$$R^2_p = 1 - \frac{l_F}{l_I}, \qquad (10)$$

where $l_F$ and $l_I$ represent the log-likelihood for the full and intercept-only models, respectively. Like $R^2$, the interpretation of $R^2_p$ is not straightforward. A lack of fit of a model is indicated by low values of $R^2_p$, whereas high values do not. However, there is no specific boundary to differentiate low and high values [19]. For the count model, incidence rate ratio (IRR) can be preferred over estimated regression coefficients to examine the effects of explanatory variables as far as interpretation and understanding are concerned. The IRR for the $k^{th}$ covariate is defined as $IRR_k = e^{\beta_k}$, $k = 1, 2, \cdots, p$.

## Results

By including climate variables, such as mean, maximum, and minimum temperatures and precipitation, as well as other variables like population demographics and population density, you can account for potential sources of variation in dengue incidence (DI). The presentation of descriptive statistics of these variables, such as mean, median, standard deviation, etc., provides an overview of the distribution of the data and helps to identify any patterns or trends in the data. The table is likely to provide a useful starting point for the analysis and modeling of the data.

In this study, climate variables, such as mean, maximum, and minimum temperatures and precipitation, as well as other variables like population demographics and population density are being considered. The presentation of descriptive statistics of these variables, such as mean, median, standard deviation, etc., provide an overview of the distribution of the data and helps to identify any patterns or trends in the data as shown in Table 1. The average dengue incidence of 129038 cases is likely a sum of cases across all 11 countries, while the mean temperature ($T_{mean}$) of 24.75°C, minimum temperature ($T_{min}$) of 19.96°C, and maximum temperature ($T_{max}$) of 29.58°C give an indication of the average climate conditions in the study region. The population density of 263.73 people per square kilometer provides a measure of the degree of crowding in the study area and can help to identify areas with a high potential for the spread of infectious diseases like dengue.

Table 2 of the study presents the average number of dengue incidences per year over the periods 2000 to 2006, 2007 to 2014, and 2015 to 2021 as well as the percentage of increase or

**Table 1. Summary statistics of dengue infected cases and explanatory variables over the years 2000–2021.**

| Variable | Minimum | Maximum | Mean | $Q_1$ | $Q_2$ | $Q_3$ |
|---|---|---|---|---|---|---|
| Dengue Infected Cases | 0 | 1649008 | 129038 | 11745 | 47596 | 147701 |
| Mean temperature (˚C) | 13.97 | 27.72 | 24.75 | 24.80 | 25.83 | 26.39 |
| Min temperature (˚C) | 7.84 | 24.11 | 19.96 | 19.14 | 21.25 | 22.05 |
| Max temperature (˚C) | 20.08 | 32.93 | 29.58 | 29.43 | 30.57 | 30.96 |
| log (*Precipitation*) | 6.594 | 8.183 | 7.557 | 7.377 | 7.538 | 7.829 |
| log (*Total population*) | 16.75 | 21.06 | 18.33 | 17.24 | 18.25 | 19.04 |
| log (*Urban*) | 14.98 | 20.02 | 17.32 | 16.55 | 17.29 | 18.45 |
| log (*Rural*) | 15.80 | 20.62 | 17.71 | 16.93 | 17.44 | 18.44 |
| Population density | 21.00 | 1163.96 | 263.73 | 95.83 | 179.18 | 324.25 |

decrease of dengue cases over the period 2015 to 2021 compared to the period 2000 to 2006. The study states that in the time period of 2000 to 2006, 2007 to 2014, and 2015 to 2021, the highest and lowest number of yearly average dengue cases are observed in Brazil and Nepal respectively. The study notes that Nepal showed a particularly high increase in dengue incidence compared to any other country, as high as 100%. Furthermore, India, Sri Lanka, and Bangladesh have also shown significant increases in dengue cases, with 95%, 88%, and 86% increases respectively. This is considered to be alarming for the South Asia region and the study suggests that global climate change is highly considered for such dengue outbreaks in South Asia. Additionally, the study finds that the percentage of dengue incidence rate is decreasing in Myanmar and Vietnam over time, however, the yearly average incidence is still extremely high in these countries. This is a concern for people living in those areas.

Exploratory data analysis (EDA) is an important step in the analysis of data and can help to identify patterns, trends, and relationships in the data. The creation of a correlation plot that includes various elements such as histograms, density functions, smoothed regression lines, correlation coefficients, and their corresponding significance levels can provide useful information regarding the connections between the response variable and predictor variables. The EDA can also help to identify any outliers or anomalies in the data that may need to be further investigated or removed from the analysis. Fig 1 provides specific information on the infected cases, mean temperature, minimum temperature, maximum temperature, log(*Precipitation*), log(*Total population*), log(*Urban*), log(*Rural*), and population density variables, along with their corresponding significance levels. From the EDA, it is observed that the distribution

**Table 2. The average number of dengue cases/year and % of the increase in the period 2015 to 2021 compared to the period 2000 to 2006.**

| Country | 2000 to 2006 | 2007 to 2014 | 2015 to 2021 | % of Increase (approx.) |
|---|---|---|---|---|
| Bangladesh | 2673 | 832 | 19,082 | 86 |
| Nepal | 04 | 286 | 2956 | 100 |
| Sri Lanka | 7298 | 29,413 | 60,520 | 88 |
| India | 5979 | 63,822 | 126,306 | 95 |
| Brazil | 246,044 | 743,077 | 956,892 | 74 |
| Indonesia | 97,402 | 144,413 | 102,949 | 5 |
| Malaysia | 24,363 | 35,000 | 55,881 | 56 |
| Philippines | 175,266 | 170,993 | 201,294 | 13 |
| Thailand | 52,617 | 53,730 | 61,481 | 14 |
| Myanmar | 20,660 | 20,474 | 19,587 | -5 |
| Vietnam | 105,586 | 195,124 | 93,895 | -12 |

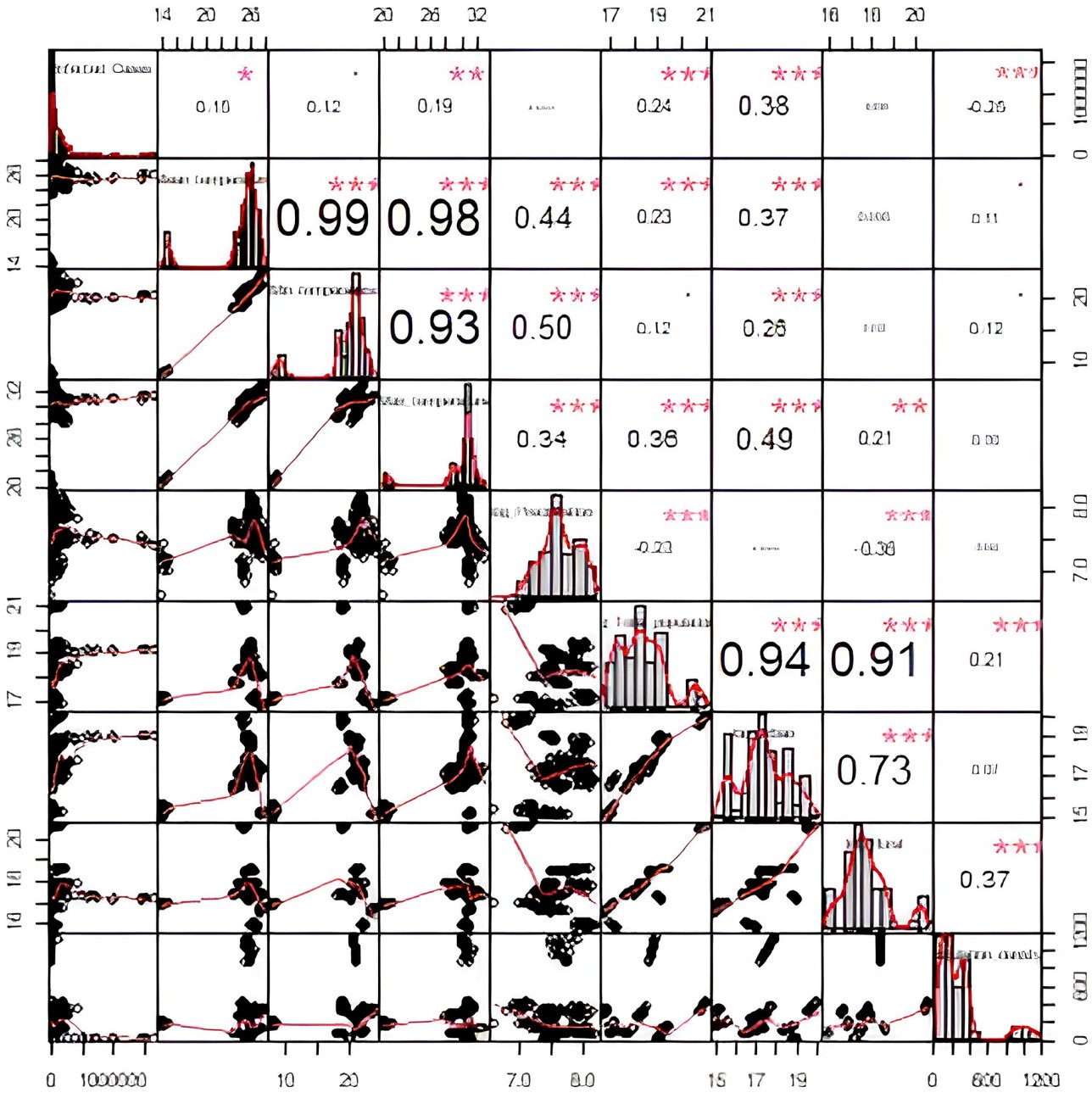

**Fig 1. Pairwise correlation plot.** (Note: p < 0.1*, p < 0.05**, p < 0.01**).

(shape) of infected cases is negatively skewed while shape of the densities of the other explanatory variables are approximately bimodal (diagonal elements of Fig 1). Furthermore, negative effects of log(*Precipitation*), log(*Rural*), and population density on infected cases can be seen in simple smooth regression lines of infected cases on log(*Precipitation*), infected cases on log(*Rural*), and infected cases on population density respectively while other explanatory variables considered in this study have positive effects on infected cases. The relationship between infected cases and each explanatory variable observed in simple smoothed regression lines provides a rough justification about the potential explanatory variables in GLM but does not

**Table 3. Goodness-of-fit tests based on AIC, Pearson residual $\chi^2$ statistic, dispersion, deviance and *Pseudo-R²* values for PR and NBR models.**

| Model | AIC | DF | Pearson $\chi^2$ | Dispersion | Deviance | $p$ – value | *Pseudo-R²* |
|---|---|---|---|---|---|---|---|
| PR | 69984.23 | 233 | 17641779 | 75715.79 | 16933275 | 0 | 0.710 |
| NBR | 23.75 | 233 | 285.937 | 1.22 | 288.65 | 0.0076 | 0.556 |

*Dispersion $= \frac{Pearson\ \chi^2}{DF}$

provide any conclusive decision about the statistical significance of explanatory variables. Furthermore, a very strong positive Pearson correlation coefficient can be seen between $T_{mean}$ and $T_{min}$, $T_{mean}$ and $T_{max}$ while Tmean has a moderate positive Pearson correlation with log(*Precipitation*), log(*Total population*), log(*Urban*). It was also noticeable that high positive correlation exists between the pairs (*Min temperature*, *Max temperature*), (log(*Total population*), log(*Urban*)), (log(*Total population*), log(*Rural*)). The existence of a strong correlation among predictors does not pose any hindrance to the application of GLM.

From the summary statistics presented in Table 1, it is observed that variance of the response variable (dengue infected cases) is 485002 times compared to its mean which indicates that the response variable is overdispersed. Therefore, overdispersion needs to be taken into account in the statistical modeling of dengue-infected cases. In this study, Poisson and negative binomial regression models were considered to analyze overdispersed dengue-infected cases, and different performance criteria of both the fitted models are presented in Table 3.

From Table 3, it is observed that dispersion value for NBR model is 1.22 while it is 75715.79 for PR. Dispersion value close to 1 is desirable for any count regression model, and this happens when the count regression model captures overdispersion sufficiently in case of overdispersed count response modeling. Hence, it can be concluded that NBR model captured overdispersion very well compared to the PR model. Furthermore, the AIC and deviance values for PR and NBR model are 69984.23 and 16933275, and 23.75 and 288.65 respectively. Therefore, NBR model can be considered as a better model to analyze dengue-infected cases compared to PR model as far as AIC and deviance values are concerned. Interestingly, pseudo-$R^2$ value (0.71) for PR is higher than pseudo-$R^2$ value (0.556) obtained for NBR model. This (misleading results) could have happened as overdispersion was not properly addressed in the PR model.

Finally, NBR model was chosen as a better model to analyze dengue-infected cases over PR model based on the above performance comparison of PR and NBR models. Therefore, only regression outputs obtained from NBR model were considered to determine the significant predictors of dengue-infected cases which are presented in Table 4.

From Table 4, it is observed that predictor variables such as $T_{mean}$ and $T_{max}$, log(*Precipitation*), log(*Urban*), and population density have statistically significant effects on dengue incidence rate while $T_{min}$, log(*Rural*), and log(T*otal population*) do not have statistically significant effects on dengue incidence rate. More specifically, the effects of log(*Precipitation*) and population density on dengue incidence rate are statistically significant at 0.1% level ($p <$ .001) while the effects of $T_{max}$ and log(*Urban*) on dengue incidence rate and the effect of $T_{mean}$ on dengue incidence rate are statistically significant at 0.6% and 0.3% levels ($p <$ .0055, $p <$ .0012) and 5% level ($p <$ .0441) respectively. The quantity of IRR presented in column 4 of Table 4 quantifies the direction of the effect of significant predictors on dengue incidence rate numerically. For example, for a one-unit increase in $T_{mean}$, the rate of dengue-infected cases increases by 2710%, with the remaining significant predictor variables values held constant.

**Table 4. Results obtained from best NBR to analyze overdispersed dengue count data collected from 11 countries in Asia.**

| Variable | Estimate | SE | IRR | 95% CI for IRR | p-value |
|---|---|---|---|---|---|
| Intercept | 17.301 | 5.006 | 3.26e+07 | (597.01, 2.32e+12) | 0.0005*** |
| *Mean temperature* | 3.336 | 1.657 | 2.81e+01 | (0.1660, 2.32e+03) | 0.0441* |
| Min temperature | -0.926 | 0.825 | 3.95e-01 | (0.0427, 5.15e+00) | 0.2619 |
| *Max temperature* | -2.334 | 0.842 | 9.68e-02 | (0.0107, 1.27e+00) | 0.0055** |
| *log(Precipitation)* | -1.648 | 0.398 | 1.92e-01 | (0.0785, 4.64e-01) | <0.001*** |
| log (Total population) | -0.721 | 0.939 | 4.86e-01 | (0.0632, 3.52e+00) | 0.4429 |
| *log(Urban)* | 1.690 | 0.523 | 5.42e+00 | (1.7526, 1.72e+01) | 0.0012** |
| log (Rural) | -0.252 | 0.397 | 7.76e-01 | (0.3506, 1.73e+00) | 0.5252 |
| *Population density* | -0.002 | 0.0003 | 9.97e-01 | (0.9970, 9.98e-01) | <0.001*** |

*p<0.05,

**p<0.01,

***p<0.001

Like $T_{mean}$, log(*Urban*) has a positive effect on dengue infected cases which is increased by 442% due to one unit increase in log(*Urban*) provided that other significant predictor variables values held constant.

On the other hand, the predictor variables $T_{max}$, log(*Precipitation*) and population density have negative effects on dengue incidence rate. For example, the expected number of dengue-infected cases decreases by 90.32% due to one unit increase in $T_{max}$ when the other significant predictors are held fixed. Similarly, the dengue cases are expected to decrease by 80.8% for a one-unit increase in log(*Precipitation*) provided that other significant predictor variables are kept fixed. Finally, for each additional unit of population density, dengue cases are expected to decrease by 0.3% when other significant predictor variables are kept fixed.

Various perspectives can be used to assess the goodness of fit of a statistical model to a given dataset. For instance, Tables 3 and 4 illustrate how NBR model compares to PR model in representing the dengue dataset. Model efficiency criteria such as AIC, dispersion, deviance, *p*-value, and *Pseudo-R²* values are used for this purpose. However, the aforementioned analysis lacks exploration regarding whether the chosen model assumptions are satisfied and whether particular data points have an impact on parameter estimates. To address these issues, diagnostic plots for generalized linear models are considered, as presented in Figs 2 and 3.

Fig 2 illustrates the jackknife deviance residuals against the linear predictor in the top left panel and the normal QQ plots of the standardized deviance residuals in the top right panel. A well-fitted model is expected to have jackknife deviance residuals distributed evenly around a horizontal line without any distinct pattern, and standardized deviance residuals lined up well on the dotted line with an intercept of zero and a slope of one. The top left plot of Fig 2 indicates that jackknife deviance residuals are distributed roughly evenly around the horizontal line y = 0. However, in the top right plot, although most of the standardized deviance residuals fall on the dotted line, some deviate from it in both the left and right-tailed areas. Specifically, the distribution of standardized deviance residuals (sample quintiles) has lighter and heavier tails in the left and right-tailed areas, respectively, compared to the theoretical standard normal distribution. This symptom does not necessarily indicate a violation of the normality assumption, but observations for which standardized deviance residuals deviate from the dotted line may be a potential problem that requires further exploration.

The bottom left and right panels of Fig 2 display Cook statistics against standardized leverages and case number, respectively. The Cook statistic measures how much the parameter

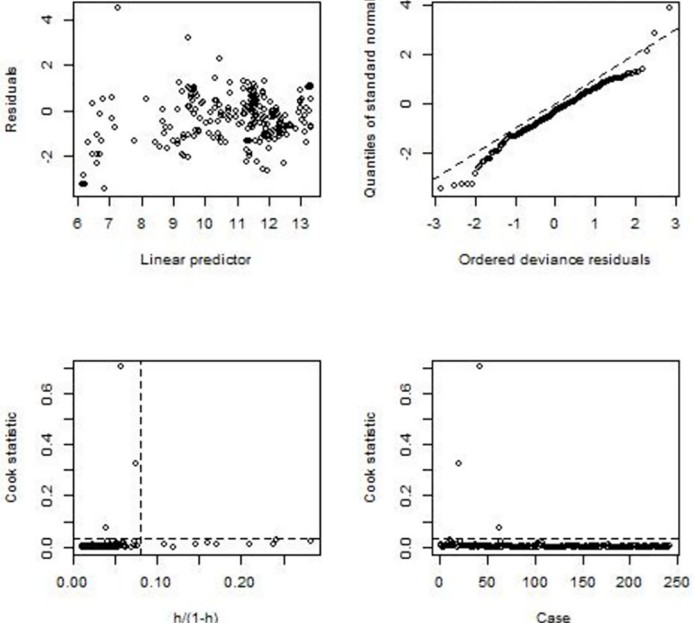

**Fig 2. Diagnostic plots for NBR model.**

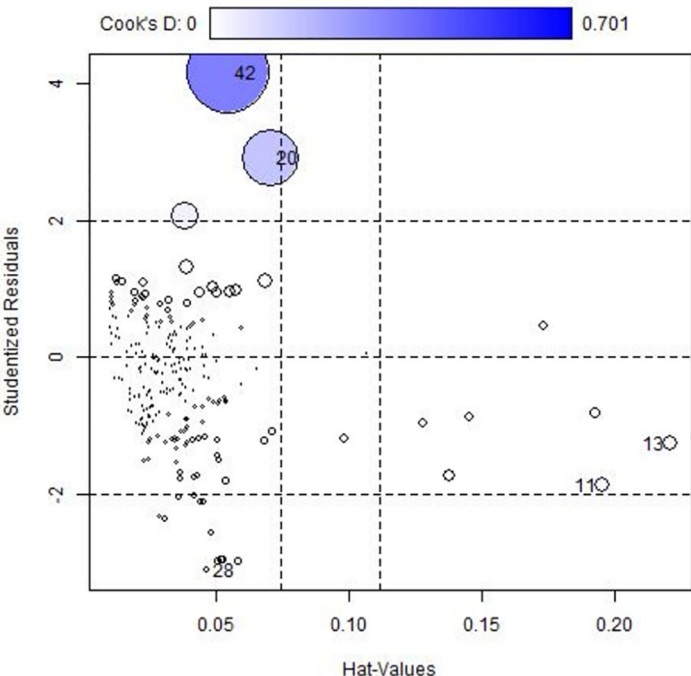

**Fig 3. Influence plot for NBR model.**

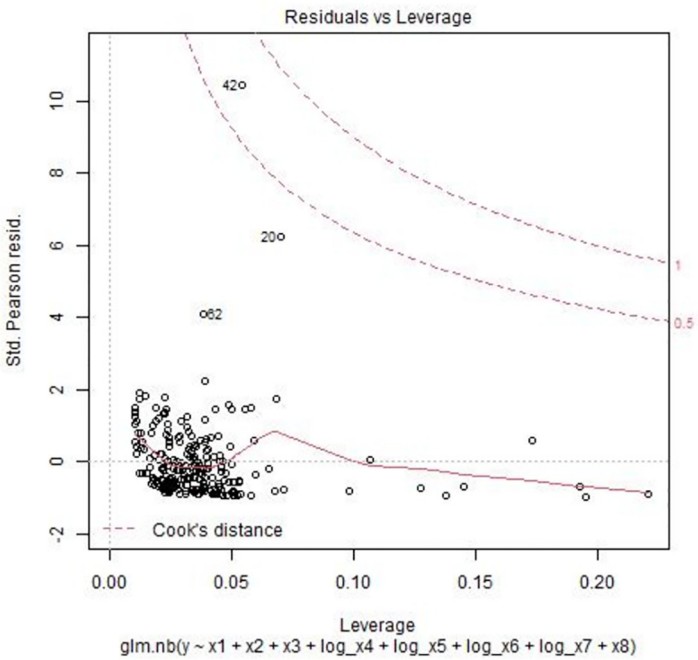

**Fig 4. Residuals vs. Leverage plot for the negative binomial model.**

estimates change when a data point is removed from the calculation of fitting the regression model. Consequently, it is employed to identify influential observations. In the bottom left graph, two dotted lines are included at $x = 2p/(n - 2p)$, and $y = 8/(n - 2p)$, where $n$ represents the count of observations and p denotes the number of estimated parameters. Points above the horizontal line (shown in the right plot) may have a high influence on the model, while points to the right of the vertical line are considered high-leverage points.

The influence plot in Fig 3 reveals that observations 20 and 42 have large positive residuals, while observation 28 has a large negative residual. Additionally, observations 11 and 13 are identified as large leverage points. The combination of a data point's leverage and residual determines its influence on a model, which can be studied using a standardized residual versus leverage plot.

The residual versus leverage plot in Fig 4 shows that observation 42 is located outside the Cook's distance in the upper right corner (marked by a red dotted line), indicating it as an influential observation. Cook's distance lines (red dotted lines) are barely visible in the residual versus leverage plot when there are no influential observations in the data. To quantify the effect of this influential observation on the model, the negative binomial regression model is fitted without observation 42. The output of this model shows that there are very slight changes in the regression coefficients, which can be ignored.

## Discussion

The study was carried out in 11 countries, and it has led to the identification of several factors that are associated with DI. The findings of the study suggest that particular environmental and demographic factors are essential in determining the occurrence of dengue. It is found that mean temperature, maximum temperature, urban population, precipitation, and population density are all associated with the incidence of dengue. In contrast, the study found that

minimum temperature, total population, and rural population had no significant association with dengue incidence. Overall, the study highlights the complex interplay between environmental and demographic factors in determining the incidence of dengue. By identifying these key factors, policymakers and public health officials can develop targeted interventions to prevent and control the spread of dengue in high-risk areas.

Aedes aegypti mosquitoes thrive in urban areas and are often found in and around human dwellings, making it easier for them to bite and infect people. When a mosquito bites an infected person, it can then become a carrier of the virus and spread the disease to other individuals it bites [21]. Urban areas are particularly vulnerable to dengue transmission due to the high densities of people and mosquitoes [22, 23]. The close proximity of people in urban areas means that the disease can spread rapidly from person to person, and the abundance of Aedes mosquitoes provides ample opportunities for the virus to be transmitted. In addition, urban areas have a high number of potential breeding sites for Aedes mosquitoes [22, 23]. These breeding sites can include anything that holds stagnant water, such as discarded tires, buckets, and other containers. When stagnant water is present, it creates favorable conditions for Aedes mosquitoes to deposit their eggs, leading to a greater number of mosquitoes and an elevated probability of dengue transmission [22, 23].

According to a study by Kolimenakis et al. [22], there is a robust association between the prevalence of dengue fever and urbanization in various countries. The investigation revealed that urban regions have around 63.3% of possible breeding sites for mosquitoes, thereby elevating the chances of dengue transmission. Additional research has also established a direct relationship between urbanization and the incidence of dengue fever [24, 25]. These findings imply that urbanization plays a crucial role as a risk factor in the propagation of the disease. In our study, we also observed that an increase of one unit in *log(Urban)* results in a 442% increase in the number of dengue-infected cases, indicating a positive impact of urbanization. While it can be challenging to restrict the movement of people from rural areas to urban centers, the limited availability of facilities and resources in rural areas can contribute to the high density of urban populations. This further exacerbates the risk of dengue transmission, as it creates more opportunities for infected individuals to come into contact with Aedes mosquitoes and facilitate the spread of the disease. Overall, the combination of urbanization, mosquitoes, and the presence of potential breeding sites in urban areas creates a significant risk for the spread of dengue fever in urban populations and across various countries. Effective measures, such as mosquito control programs and public awareness campaigns, are necessary to prevent and control the spread of dengue fever in urban areas.

The Aedes mosquito thrives in warm and humid environments [26], with optimal temperatures for their development ranging from 26 to 32˚C [26]. The rise in mosquito populations during the warm and wet seasons is known to contribute to an increase in dengue fever incidence [26]. Additionally, higher temperatures can speed the virus's development within mosquitoes, resulting in a shorter incubation period and a higher chance of transmission [26]. The present study further supports this relationship, demonstrating a positive correlation between mean temperature and dengue fever incidence. Specifically, a one-unit increase in mean temperature is linked to a significant increase of 2710% in the rate of dengue cases, holding all other significant predictor variables constant. Research conducted in several regions of the world [27–29] has indicated that there is a direct relationship between the $T_{mean}$ and DI. This evidence suggests that as temperatures increase due to climate change, there may be a corresponding increase in dengue incidence in various parts of the world.

Heavy rainfall, particularly precipitation, can play a vital role in decreasing dengue incidence by flushing out mosquito breeding sites and larvae. When heavy rainfall occurs, it washes away standing water and disrupts breeding sites, ultimately reducing the mosquito

population and decreasing the likelihood of dengue transmission. Additionally, heavy rainfall helps to decrease the ambient temperature, further reducing the mosquito population and the risk of dengue transmission. Various research studies including Roiz et al. [30], Dieng et al. [31], and Romiti et al. [32] have explained that precipitation has a negative effect on dengue spreading due to a decrease in the number of mosquito eggs. Our research also supports the concept that heavy rainfall can have a positive impact on reducing dengue incidence. We observed a negative correlation between *log(precipitation)* and the dengue incidence rate, indicating that as precipitation levels increase, the incidence of dengue decreases. Therefore, heavy rainfall may be a significant factor in reducing the incidence of dengue, as previously discussed.

Numerous studies suggest a positive correlation between population density and dengue fever incidence, though this relationship is dependent on various factors. However, Schmidt et al. [33] found that regions with high population densities did not experience severe dengue outbreaks and had a lower risk of such outbreaks. Araujoa et al. [34] also observed that higher population density regions had a lower incidence of dengue compared to low population density regions. Furthermore, Istiqamah et al. [35] reported that there is no significant association between population density and dengue incidence, suggesting that the severity of dengue outbreaks is not necessarily linked to higher human population densities. Our study reveals findings that are consistent with the previously cited studies, demonstrating a negative correlation between population density and dengue incidence. This suggests that regions with higher population densities may not necessarily experience more severe dengue outbreaks. Furthermore, the negative correlation observed between population density and dengue incidence in our study could be attributed to several factors. For instance, higher population densities may lead to greater awareness and access to healthcare resources, resulting in improved diagnosis and treatment of dengue cases [36]. Additionally, densely populated areas may be more likely to have effective vector control measures in place, which can help to limit the spread of dengue.

## Conclusion

In this study, we analyzed data from 11 dengue-prone Asian countries between 2000 and 2021 to identify the environmental and demographic factors associated with dengue infection. The analysis revealed the following:

- The variance of dengue cases was 485002 times greater than its mean, indicating that the data was overdispersed.

- NBR model fit the data better than the PR model. Therefore, the significant factors affecting dengue cases were determined using the outputs obtained from the negative binomial regression model.

- $T_{mean}$, $T_{max}$, log(*Precipitation*), log(*Urban*), and population density have statistically significant effects on dengue incidence rate.

## Limitations

Our study's findings are largely consistent with existing literature, except for the results related to population density. We attribute this inconsistency to the unavailability of information regarding factors such as individuals' immunity levels, degree of awareness, and effectiveness of vector control measures in the study area, all of which are believed to influence the incidence of dengue. Unfortunately, these variables were not included in our dataset, limiting the

scope of our study. To obtain a more comprehensive understanding of the association between population density and dengue incidence, future research could overcome this limitation by considering these factors in their analysis.

There are numerous variables that cannot be analyzed because it is likely that this information is not available. Our study solely relies on secondary information, which may introduce biases in data collection. Only population-level information is analyzed, while individual-level data has not been utilized. Consequently, it is impossible to separate the biological factors that can influence the disease, as these factors cannot be included in this analysis.

## Supporting information

**S1 Appendix.**
(DOCX)

## Author Contributions

**Conceptualization:** Goutam Saha.

**Data curation:** Anamul Haque Sajib, Sabina Akter, Goutam Saha.

**Formal analysis:** Anamul Haque Sajib, Goutam Saha.

**Funding acquisition:** Goutam Saha.

**Investigation:** Anamul Haque Sajib, Goutam Saha.

**Methodology:** Anamul Haque Sajib, Goutam Saha.

**Project administration:** Goutam Saha.

**Resources:** Goutam Saha.

**Supervision:** Goutam Saha.

**Validation:** Anamul Haque Sajib, Sabina Akter, Goutam Saha.

**Visualization:** Anamul Haque Sajib, Sabina Akter, Goutam Saha.

**Writing – original draft:** Anamul Haque Sajib, Sabina Akter, Goutam Saha, Zakir Hossain.

**Writing – review & editing:** Anamul Haque Sajib, Goutam Saha, Zakir Hossain.

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
