## [Decision Letter · Decision Letter 0]

2 May 2024

PONE-D-24-11200Demographic-Environmental Effect on Dengue Outbreaks in 11 CountriesPLOS ONE

Dear Dr. Saha,

Thank you for submitting your manuscript to PLOS ONE. After careful consideration, we feel that it has merit but does not fully meet PLOS ONE’s publication criteria as it currently stands. Therefore, we invite you to submit a revised version of the manuscript that addresses the points raised during the review process. While both reviewers felt the manuscript had solid scientific merit, there were a number of concerns and recommendations that must be addressed before the manuscript can be further considered for publication. In particular, reviewer 2 raised important concerns about the justification of the selection of countries for the study, as well as the climate variables included in the analyses; please address these in subsequent drafts. Additionally please ensure to address comments about recommendations for modifications for the Introduction, Methodology, Results, and Discussion raised by both reviewers.

We look forward to receiving your revised manuscript.

Kind regards,

James Colborn

Academic Editor

PLOS ONE

Journal Requirements:

Reviewers' comments:

Reviewer's Responses to Questions

**Comments to the Author**

1. Is the manuscript technically sound, and do the data support the conclusions?

Reviewer #1: Yes

Reviewer #2: Partly

2. Has the statistical analysis been performed appropriately and rigorously? 

Reviewer #1: Yes

Reviewer #2: Yes

3. Have the authors made all data underlying the findings in their manuscript fully available?

Reviewer #1: Yes

Reviewer #2: Yes

4. Is the manuscript presented in an intelligible fashion and written in standard English?

Reviewer #1: Yes

Reviewer #2: No

5. Review Comments to the Author

Reviewer #1: I would like to congratulate the authors for the effort in writing this paper, I think it is important to continue research in this field and use safe methods to improve the quality of research.

I was reviewing this paper and I really found some observations, but I think it is important that they take these observations into account to improve the information they want to communicate.

Introduction is good

Methods is fine

Results is OK

Discussion: In this part you need to improve the discussion with more bibliography in relation to the information included in the text for example:

Bibliography should be included in the following paragraph, because there is information that seems obvious but is not so obvious, especially on biology and transmission of Aedes:

“Aedes aegypti mosquitoes thrive in urban areas and are often found in and around human dwellings, making it easier for them to bite and infect people. When a mosquito bites an infected person, it can then become a carrier of the virus and spread the disease to other individuals it bites.[Bibliography] Urban areas are particularly vulnerable to dengue transmission due to the high densities of people and mosquitoes. [Bibliography] The close proximity of people in urban áreas means that the disease can spread rapidly from person to person, and the abundance of Aedes mosquitoes provides ample opportunities for the virus to be transmitted. In addition, urban areas have a high number of potential breeding sites for Aedes mosquitoes. [Bibliography] These breeding sites can include anything that holds stagnant water, such as discarded tires, buckets, and other containers. When stagnant water is present, it creates favorable conditions for Aedes mosquitoes to deposit their eggs, leading to a greater number of mosquitoes and an elevated probability of dengue transmission.” [Bibliography]

In an other pargraph you should include bibliography:

“The Aedes mosquito thrives in warm and humid environments, [Bibliography] with optimal temperaturas for their development ranging from 26 to 32 °C. [Bibliography] The rise in mosquito populations during the warm and wet seasons is known to contribute to an increase in dengue fever incidence. [Bibliography]

Additionally, higher temperatures can hasten the virus's development within mosquitoes, resulting in a shorter incubation period and a higher chance of transmission. [Bibliography]”

Please include literature on this topic

It is important to be honest in the description of the limitations of the study, regarding this topic you should include a limitation on ecological studies, all of them have limitations such as there are many variables that cannot be analysed because it is likely that this information is not found, in your study only secondary information is used and this information could have biases in the collection of information, only population information is analysed and individual information has not been used and it is impossible to know the biological factors that can influence the disease, as these factors cannot be included in this analysis.

Reviewer #2: Introduction:

- What is the research gap? Based on the research gap, what are the study objectives? What is the significance of this study? The author must create a new paragraph after the last paragraph of the “Introduction” section and incorporate these items.

Data Description:

- Provide a detailed justification for the selection of 11 countries.

- Adding a few more climate variables to the study would be better.

- Since the number of yearly dengue cases are collected from the “Website of Ministry of Health” of the 11 countries, it is better to provide all the website links in the appendix section.

Methodology:

- Provide more justification for using the PR and NBR model.

Results:

- Why are the logarithms used for Precipitation, Total population, Urban and Rural variables?

- Need to improve the qualities of all four figures.

6. PLOS authors have the option to publish the peer review history of their article (what does this mean?). If published, this will include your full peer review and any attached files.

Reviewer #1: **Yes: **Dante R Culqui L

Reviewer #2: No

---

## [Author Response · Author response to Decision Letter 0]

3 May 2024

PONE-D-24-11200

Demographic-Environmental Effect on Dengue Outbreaks in 11 Countries

We like to thank all the reviewers for their valuable comments and feedback. It helps us to improve the manuscript.

Answer to the Reviewers' comments:

Q1. Is the manuscript technically sound, and do the data support the conclusions?

Reviewer #1: Yes

Reviewer #2: Partly

Answer: Thank you very much.

Q2. Has the statistical analysis been performed appropriately and rigorously?

Reviewer #1: Yes

Reviewer #2: Yes

Answer: Thank you very much for supporting our statistical analysis. 

Q3. Have the authors made all data underlying the findings in their manuscript fully available?

Reviewer #1: Yes

Reviewer #2: Yes

Answer: Thank you very much. We made available our data for all. 

Q4. Is the manuscript presented in an intelligible fashion and written in Standard English?

Reviewer #1: Yes

Reviewer #2: No

Answer: Thank you very much. We will check the manuscript again and will try to remove all the grammatical errors. 

Reviewer #1:

I would like to congratulate the authors for the effort in writing this paper, I think it is important to continue research in this field and use safe methods to improve the quality of research. I was reviewing this paper and I really found some observations, but I think it is important that they take these observations into account to improve the information they want to communicate.

Introduction is good, Methods is fine, and Results is OK.

Thank you very much. We are really grateful to you for such encouraging words. 

Q5. Discussion: In this part you need to improve the discussion with more bibliography in relation to the information included in the text for example: Bibliography should be included in the following paragraph, because there is information that seems obvious but is not so obvious, especially on biology and transmission of Aedes:

“Aedes aegypti mosquitoes thrive in urban areas and are often found in and around human dwellings, making it easier for them to bite and infect people. When a mosquito bites an infected person, it can then become a carrier of the virus and spread the disease to other individuals it bites. [Bibliography] Urban areas are particularly vulnerable to dengue transmission due to the high densities of people and mosquitoes. [Bibliography] The close proximity of people in urban areas means that the disease can spread rapidly from person to person, and the abundance of Aedes mosquitoes provides ample opportunities for the virus to be transmitted. In addition, urban areas have a high number of potential breeding sites for Aedes mosquitoes. [Bibliography] These breeding sites can include anything that holds stagnant water, such as discarded tires, buckets, and other containers. When stagnant water is present, it creates favorable conditions for Aedes mosquitoes to deposit their eggs, leading to a greater number of mosquitoes and an elevated probability of dengue transmission.” [Bibliography]

Answer: Thank you very much. 

Aedes aegypti mosquitoes thrive in urban areas and are often found in and around human dwellings, making it easier for them to bite and infect people. When a mosquito bites an infected person, it can then become a carrier of the virus and spread the disease to other individuals it bites [33]. Urban areas are particularly vulnerable to dengue transmission due to the high densities of people and mosquitoes [21, 35]. The close proximity of people in urban areas means that the disease can spread rapidly from person to person, and the abundance of Aedes mosquitoes provides ample opportunities for the virus to be transmitted. In addition, urban areas have a high number of potential breeding sites for Aedes mosquitoes [21, 35]. These breeding sites can include anything that holds stagnant water, such as discarded tires, buckets, and other containers. When stagnant water is present, it creates favorable conditions for Aedes mosquitoes to deposit their eggs, leading to a greater number of mosquitoes and an elevated probability of dengue transmission [21, 35].

Q6. In another paragraph you should include bibliography:

“The Aedes mosquito thrives in warm and humid environments, [Bibliography] with optimal temperaturas for their development ranging from 26 to 32 °C. [Bibliography] The rise in mosquito populations during the warm and wet seasons is known to contribute to an increase in dengue fever incidence. [Bibliography] Additionally, higher temperatures can hasten the virus's development within mosquitoes, resulting in a shorter incubation period and a higher chance of transmission. [Bibliography]”

Answer: Thank you very much.

The Aedes mosquito thrives in warm and humid environments [34], with optimal temperatures for their development ranging from 26 to 32 °C [34]. The rise in mosquito populations during the warm and wet seasons is known to contribute to an increase in dengue fever incidence [34]. Additionally, higher temperatures can speed the virus's development within mosquitoes, resulting in a shorter incubation period and a higher chance of transmission [34].

Q7. Please include literature on this topic

It is important to be honest in the description of the limitations of the study, regarding this topic you should include a limitation on ecological studies, all of them have limitations such as there are many variables that cannot be analysed because it is likely that this information is not found, in your study only secondary information is used and this information could have biases in the collection of information, only population information is analysed and individual information has not been used and it is impossible to know the biological factors that can influence the disease, as these factors cannot be included in this analysis.

Answer: Thank you very much. We added the following statement in our limitation section:

There are numerous variables that cannot be analyzed because it is likely that this information is not available. Our study solely relies on secondary information, which may introduce biases in data collection. Only population-level information is analyzed, while individual-level data has not been utilized. Consequently, it is impossible to separate the biological factors that can influence the disease, as these factors cannot be included in this analysis.

Reviewer #2: 

Q8. Introduction: - What is the research gap? Based on the research gap, what are the study objectives? What is the significance of this study? The author must create a new paragraph after the last paragraph of the “Introduction” section and incorporate these items.

Answer: Thank you very much. Please see the following updated paragraph:

While numerous studies have investigated the association between climate variables and dengue transmission, there is a noticeable gap in research that integrates demographic and environmental factors into the analysis, particularly in highly affected countries. Despite many literature on climate-dengue relationships, there remains a need for comprehensive studies that consider the relation between socioeconomic variables, environmental conditions, and dengue spread. Considering the research gap, this research aims to understand the relationship between demographic-environmental factors and the spread of dengue transmission, with a focus on countries that are highly affected by dengue. The use of statistical models and graphical techniques allows for a comprehensive analysis of the data to identify the most influential factors and determine the strength of their relationship with dengue transmission. This study holds significant importance, firstly, by examining the relationship between demographic-environmental factors and dengue transmission in highly affected countries, it addresses a critical gap in the existing literature, providing insights of dengue outbreaks. Secondly, the utilization of statistical models and graphical techniques enables a comprehensive analysis of the data, facilitating the identification of key factors and their importance in driving dengue transmission.

Q9. Data Description: 

(a) Provide a detailed justification for the selection of 11 countries.

(b) Adding a few more climate variables to the study would be better.

(c) Since the number of yearly dengue cases are collected from the “Website of Ministry of Health” of the 11 countries, it is better to provide all the website links in the appendix section.

Answer: Thank you very much.

(a) The selection of Bangladesh, Nepal, Sri Lanka, India, Brazil, Indonesia, Malaysia, Philippines, Thailand, Myanmar, and Vietnam for data collection in this study is justified by their significant burden of dengue fever, diverse geographical landscapes, and varying socioeconomic conditions. Moreover, the inclusion of these countries allows for the exploration of diverse climatic zones and population densities, thereby enabling a clear understanding of the contextual factors influencing dengue spread.

(b) We are sorry for not adding more climate variables due to lack of available data for these countries.

(c) Appendix:

Bangladesh: https://old.dghs.gov.bd/index.php/bd/

Nepal: https://mohp.gov.np/en

Sri Lanka: https://www.dengue.health.gov.lk/

India: https://ncvbdc.mohfw.gov.in/index.php

Indonesia: https://www.kemkes.go.id/eng/home

Malaysia: https://iku.moh.gov.my/

Philippines: https://doh.gov.ph/

Thailand: https://ddc.moph.go.th/viralpneumonia/eng/index.php

Myanmar: https://moh.nugmyanmar.org/

Vietnam: https://moh.gov.vn/web/ministry-of-health

Q10: Methodology: - Provide more justification for using the PR and NBR model.

Answer: Thank you very much. Please see the following updated paragraph:

The response variable considered in our study is the number of dengue cases which is a count (discrete) variable in nature. We aim to explore how this response variable is affected by demographic and environmental factors, and this type of relationship can be statistically examined through several count regression models, most importantly Poisson, Negative binomial, zero inflated Poisson and zero inflated Negative binomial models under generalized linear model framework. For a particular data set, suitable count regression models are selected based on the relationship of response variable mean and variance. For example, (i) Poisson regression can be used when mean and variance of the count response are equal, (ii) Negative binomial regression can be used in over dispersed count data (variance of the count response is larger than its mean), (iii) Zero inflated Poisson can be used when response consists of excessive zero count compare to the other counts but its mean and variance are the same and (iv) Zero inflated negative binomial can be used in case of excessive zero counts with over-dispersion. The response variable in our data set is over dispersed as its variance is 485002 times compared to its mean and no excessive zero (even other numbers) count is observed in the data. Therefore, Negative binomial regression model is chosen as a suitable count regression model to analyze the data considered in this paper. We also consider here Poisson regression model as a basic count regression model.

Q11: Results: - Why are the logarithms used for Precipitation, Total population, Urban and Rural variables?

Answer: Thank you very much. The answer of the above query was available in the paragraph under equation 7 in the original manuscript. However, an updated paragraph with a bit more details related to this query is provided in the revised manuscript. Please see the updated following paragraph:

Some of the covariates considered in our study like precipitation, total population, unban population and rural population have very higher values compared to the other covariates values such as mean temperature, min temperature, max temperature and population density. More specifically, the numerical values of mean temperature, min temperature and max temperature are limited to two digits before decimal while population density has three digits value. On the other hand, total population, urban population and rural population have values in 9 digits form while precipitation has 4 digits value. Therefore, scaling of the independent variable values differs drastically which causes numerical instability in the estimation process of the count regression model. We considered log transformation of precipitation, total population, unban population and rural population to make their values similar to other independent variables values which provides numerical stability in the estimation process. 

Q12: - Need to improve the qualities of all four figures.

Answer: Thank you very much. We tried to update the quality of these four figures.

Additional Updates:

We changed the manuscript format as PLOS ONE requirement. We also update the references.

---

## [Decision Letter · Decision Letter 1]

6 Jun 2024

Demographic-Environmental Effect on Dengue Outbreaks in 11 Countries

PONE-D-24-11200R1

Dear Dr. Saha,

We’re pleased to inform you that your manuscript has been judged scientifically suitable for publication and will be formally accepted for publication once it meets all outstanding technical requirements.

Kind regards,

James Colborn

Academic Editor

PLOS ONE

Additional Editor Comments (optional):

Reviewers' comments:

Reviewer's Responses to Questions

**Comments to the Author**

1. If the authors have adequately addressed your comments raised in a previous round of review and you feel that this manuscript is now acceptable for publication, you may indicate that here to bypass the “Comments to the Author” section, enter your conflict of interest statement in the “Confidential to Editor” section, and submit your "Accept" recommendation.

Reviewer #1: All comments have been addressed

Reviewer #2: All comments have been addressed

2. Is the manuscript technically sound, and do the data support the conclusions?

Reviewer #1: Yes

Reviewer #2: Yes

3. Has the statistical analysis been performed appropriately and rigorously? 

Reviewer #1: Yes

Reviewer #2: Yes

4. Have the authors made all data underlying the findings in their manuscript fully available?

Reviewer #1: Yes

Reviewer #2: Yes

5. Is the manuscript presented in an intelligible fashion and written in standard English?

Reviewer #1: Yes

Reviewer #2: Yes

6. Review Comments to the Author

Reviewer #1: Congratulations, I believe this study can be published, if the editors agree with my opinion.

I have no further comments on your article.

Kind regards

Reviewer #2: (No Response)

7. PLOS authors have the option to publish the peer review history of their article (what does this mean?). If published, this will include your full peer review and any attached files.

Reviewer #1: **Yes: **Dante R Culqui Lévano (https://orcid.org/0000-0003-1570-8012)

Reviewer #2: No

---

## [Editor Report · Acceptance letter]

3 Jul 2024

PONE-D-24-11200R1 

PLOS ONE

Dear Dr. Saha, 

I'm pleased to inform you that your manuscript has been deemed suitable for publication in PLOS ONE. Congratulations! Your manuscript is now being handed over to our production team.

Kind regards, 

on behalf of

Dr. James Colborn 

Academic Editor

PLOS ONE